# Universal Approximation with Deep Narrow Networks

## Abstract

The classical Universal Approximation Theorem certifies that the universal approximation property holds for the class of neural networks of arbitrary width. Here we consider the natural 'dual' theorem for width-bounded networks of arbitrary depth. Precisely, let $n$ be the number of inputs neurons, $m$ be the number of output neurons, and let $\rho$ be any nonaffine continuous function, with a continuous nonzero derivative at some point. Then we show that the class of neural networks of arbitrary depth, width $n + m + 2$, and activation function $\rho$, exhibits the universal approximation property with respect to the uniform norm on compact subsets of $\mathbb{R}^n$. This covers every activation function possible to use in practice; in particular this includes polynomial activation functions, making this genuinely different to the classical case. We go on to consider extensions of this result. First we show an analogous result for a certain class of nowhere differentiable activation functions. Second we establish an analogous result for noncompact domains, by showing that deep narrow networks with the ReLU activation function exhibit the universal approximation property with respect to the $p$-norm on $\mathbb{R}^n$. Finally we show that width of only $n + m + 1$ suffices for 'most' activation functions.

## 1 Introduction

The Universal Approximation Theorem (Cybenko, 1989; Hornik, 1991; Pinkus, 1999) states that universal approximation holds for the class of neural networks with a single hidden layer of arbitrary width, with any continuous nonpolynomial activation function:

**Theorem 1.1.** *Let $\rho \colon \mathbb{R} \to \mathbb{R}$ be any continuous function. Let $\mathcal{N}_n^\rho$ represent the class of neural networks with activation function $\rho$, with $n$ neurons in the input layer, one neuron in the output layer, and one hidden layer with an arbitrary number of neurons. Let $K \subseteq \mathbb{R}^n$ be compact. Then $\mathcal{N}_n^\rho$ is dense in $C(K)$ if and only if $\rho$ is nonpolynomial.*

What if arbitrary width is replaced with arbitrary depth? Put more precisely, can networks of bounded width and arbitrary depth provide universal approximation? In some sense this poses a question 'dual' to the problem answered by the classical Universal Approximation Theorem. We refer to networks of this type as *deep, narrow* networks.

Furthermore we might ask how narrow the network may be, and what activation functions may be admitted. We provide a near-complete answer to these various questions.

Universal approximation may be established with respect to more than one topology. Continuous activation functions beget networks representing continuous functions. Thus when working with respect to the uniform norm, it is natural to seek density in $C(K; \mathbb{R}^m)$ for $K \subseteq \mathbb{R}^n$. When working with respect to the $p$-norm, it is natural to seek density in $L^p(\mathbb{R}^n; \mathbb{R}^m)$ for $p \in [1, \infty)$. In this latter case we may hope to generalise to noncompact domains, as functions in $L^p(\mathbb{R}^n; \mathbb{R}^m)$ must exhibit some sort of decay.

The primary motivation for this work stems from the work of Lu et al. (2017), who study this question in the special case of the popular ReLU activation function, and who establish density in $L^1(\mathbb{R}^n)$. The other notable result we are aware of is the work of Hanin & Sellke (2017), who show another special case: they also consider the ReLU activation function, and establish density in $C(K; \mathbb{R}^m)$ for $K \subseteq \mathbb{R}^n$ compact.

This article demonstrates generalisations of these results, in particular to general activation functions, without relying on the strong algebraic and analytic properties of the ReLU activation function. This also improves certain results specific to the ReLU.

The rest of the paper is laid out as follows. Section 2 discusses existing work. Section 3 provides a brief summary of our results; these are then presented in detail in Section 4. Section 5 is the conclusion. Several proofs are deferred to the appendices, due to length and technical content.

## 2 CONTEXT

Some positive results have been established showing that particular classes of networks are dense in certain spaces. Some negative results have also been established, showing that insufficiently wide networks will fail to be dense.

Hanin & Sellke (2017) have shown that deep narrow networks with the ReLU activation function exhibit the universal approximation property in $C(K; \mathbb{R}^m)$ for $K \subseteq \mathbb{R}^n$ compact.

Lu et al. (2017) have shown that deep narrow networks with the ReLU activation function exhibit the universal approximation property in $L^1(\mathbb{R}^n)$, whilst Lin & Jegelka (2018) have shown that a particular description of residual networks, with the ReLU activation function, also exhibit the universal approximation property in this space. We are not aware of any results for the general case of $L^p(\mathbb{R}^n; \mathbb{R}^m)$ for $p \in [1, \infty)$.

We do not know of any positive results applying to activation functions other than the ReLU.

Regarding widths insufficient for a class of deep narrow networks to exhibit the universal approximation property, consider the case of a network with $n$ input neurons and a single output neuron. For certain activation functions, Johnson (2019) shows that width $n$ is insufficient to give density in $C(K)$. For the ReLU activation function, Lu et al. (2017) show that width $n$ is insufficient to give density in $L^1(\mathbb{R}^n)$, and that width $n-1$ is insufficient in $L^1([-1, 1]^n)$. For the ReLU activation function, Hanin & Sellke (2017) shows that width $n$ is insufficient to give density in $C(K)$, and that in fact that this is the greatest possible width not achieving universal approximation in this context.

The precise minimum width for activation functions other than ReLU, or for multiple output neurons, remains unknown.

Everything discussed so far is in the most general case of approximating functions on Euclidean space: in the language of machine learning, they are regression tasks. There has been some related work in the special case of classification tasks, for example Beise et al. (2018); Szymanski & McCane (2012); Rojas (2003); Nguyen et al. (2018). There has also been some related work in the special case of certain finite domains; Le Roux & Bengio (2010) show that networks with sigmoid activation function and width $n$ can approximate any distribution on $\{0, 1\}^n$. See also Sutskever & Hinton (2008). Montúfar (2014) considers the analogous scenario for distributions on $\{0, 1, \ldots, q-1\}^n$.

## 3 SUMMARY OF RESULTS

**Definition 3.1.** Let $\rho \colon \mathbb{R} \to \mathbb{R}$ and $n, m, k \in \mathbb{N}$. Then let $\mathcal{NN}_{n,m,k}^\rho$ represent the class of functions $\mathbb{R}^n \to \mathbb{R}^m$ described by neural networks with $n$ neurons in the input layer, $m$ neurons in the output layer, $k$ neurons in each hidden layer, and an arbitrary number of hidden layers, such that every neuron in every hidden layer has activation function $\rho$, and every neuron in the output layer has the identity activation function.

Our central result is the following theorem.

**Theorem 3.2.** *Let $\rho \colon \mathbb{R} \to \mathbb{R}$ be any continuous function which is continuously differentiable at at least one point, with nonzero derivative at that point. Let $K \subseteq \mathbb{R}^n$ be compact. Then $\mathcal{NN}_{n,m,n+m+2}^\rho$ is dense in $C(K; \mathbb{R}^m)$.*

The technical condition is very weak; in particular it is satisfied by every piecewise-$C^1$ function not identically zero. Thus any activation function that one might practically imagine using on a computer must satisfy this property.

Theorem 3.2 is proved by handling particular classes of activation functions as special cases. First we have the result for nonpolynomial activation functions, for which the width can be made slightly smaller.

**Theorem 4.4.** *Let $\rho\colon \mathbb{R} \to \mathbb{R}$ be any continuous nonpolynomial function which is continuously differentiable at at least one point, with nonzero derivative at that point. Let $K \subseteq \mathbb{R}^n$ be compact. Then $\mathcal{NN}^\rho_{n,m,n+m+1}$ is dense in $C(K;\mathbb{R}^m)$.*

We observe a corollary for noncompact domains, which generalises Lu et al. (2017, Theorem 1) to multiple output neurons, a narrower width, and $L^p$ for $p \geqslant 1$ instead of just $p = 1$.

**Corollary 4.6.** *Let $\rho$ be the ReLU activation function. Let $p \in [1,\infty)$. Then $\mathcal{NN}^\rho_{n,m,n+m+1}$ is dense in $L^p(\mathbb{R}^n;\mathbb{R}^m)$.*

Moving on to polynomial activation functions, the smaller width of $n + m + 1$ also suffices for a large class of polynomials.

**Theorem 4.8.** *Let $\rho\colon \mathbb{R} \to \mathbb{R}$ be any polynomial for which there exists a point $\alpha \in \mathbb{R}$ such that $\rho'(\alpha) = 0$ and $\rho''(\alpha) \neq 0$. Let $K \subseteq \mathbb{R}^n$ be compact. Then $\mathcal{NN}^\rho_{n,m,n+m+1}$ is dense in $C(K;\mathbb{R}^m)$.*

The simplest example of such a $\rho$ is $x \mapsto x^2$. Note that in the classical arbitrary-width case it is both necessary and sufficient that the activation function be nonpolynomial. Here, however, the same restriction does not hold. Polynomial activation functions are a reasonable choice in this context.

The technical restrictions on the polynomial may be lifted by allowing the full $n + m + 2$ neurons per hidden layer.

**Theorem 4.10.** *Let $\rho\colon \mathbb{R} \to \mathbb{R}$ be any nonaffine polynomial. Let $K \subseteq \mathbb{R}^n$ be compact. Then $\mathcal{NN}^\rho_{n,m,n+m+2}$ is dense in $C(K;\mathbb{R}^m)$.*

It is clear that Theorems 4.4 and 4.10 together imply Theorem 3.2.

Finally we observe that even pathological cases not satisfying the technical condition of Theorem 3.2 may exhibit the universal approximation property.

**Proposition 4.13.** *Let $w\colon \mathbb{R} \to \mathbb{R}$ be any bounded continuous nowhere differentiable function. Let $\rho(x) = \sin(x) + w(x)\mathrm{e}^{-x}$, which will also be nowhere differentiable. Let $K \subseteq \mathbb{R}^n$ be compact. Then $\mathcal{NN}^\rho_{n,m,n+m+1}$ is dense in $C(K;\mathbb{R}^m)$.*

Whilst not of direct practical application, this result exemplifies that little necessarily needs to be assumed about an activation function to understand the corresponding class of neural networks.

**Remark 3.3.** Every proof in this article is constructive, and can in principle be traced so as to determine how depth changes with approximation error. We have instead chosen to focus on quantifying the width necessary for universal approximation. In fact there are places in our arguments where we have used a deeper network over a shallower one, when the deeper network is more easily explained.

**Remark 3.4.** An understanding of universal approximation in deep narrow networks is applicable to an understanding of bottlenecks, when information must be discarded due to space constraints, for example in autoencoders (Bengio et al., 2006). This article demonstrates that certain narrow networks will *not* constitute a bottleneck; a converse example is Johnson (2019), who demonstrates that networks of insufficient width are forced to maintain certain topological invariants.

## 4 Universal approximation

### 4.1 Preliminaries

**Remark 4.1.** A neuron is usually defined as an activation function composed with an affine function. For ease, we shall extend the definition of a neuron to allow it to represent a function of the form $\psi \circ \rho \circ \phi$, where $\psi$ and $\phi$ are affine functions, and $\rho$ is the activation function. This does not increase the representational power of the network, as the new affine functions may be absorbed into the affine parts of the next layer, but it will make the neural representation of many functions easier to present. We refer to these as *enhanced neurons*. It is similarly allowable to take affine combinations of multiple enhanced neurons; we will use this fact as well.

One of the key ideas behind our constructions is that most reasonable activation functions can be taken to approximate the identity function. Indeed, this is essentially the notion that differentiability captures: that a function is locally affine. This makes it possible to treat neurons as 'registers', in which information may be stored and preserved through the layers. This allows for preserving the input values between layers, which is crucial to performing computations in a memory-bounded regime. Thus our constructions have strong overtones of space-limited algorithm design in traditional computer science settings.

**Lemma 4.2.** *Let $\rho\colon \mathbb{R} \to \mathbb{R}$ be any continuous function which is continuously differentiable at at least one point, with nonzero derivative at that point. Let $L \subseteq \mathbb{R}$ be compact. Then a single enhanced neuron with activation function $\rho$ may uniformly approximate the identity function $\iota\colon \mathbb{R} \to \mathbb{R}$ on $L$, with arbitrarily small error.*

*Proof.* By assumption, as $\rho$ is *continuously* differentiable, there exists $[a,b] \subseteq \mathbb{R}$ with $a \neq b$, on some neighbourhood of which $\rho$ is differentiable, and $\alpha \in (a,b)$ at which $\rho'$ is continuous, and for which $\rho'(\alpha)$ is nonzero.

For $h \in \mathbb{R} \setminus \{0\}$, let $\phi_h(x) = hx + \alpha$, and let

$$\psi_h(x) = \frac{x - \rho(\alpha)}{h\rho'(\alpha)}.$$

Then

$$\iota_h = \psi_h \circ \rho \circ \phi_h$$

is of the form that an enhanced neuron can represent. Then for all $u \in [a,b]$, by the Mean Value Theorem there exists $\xi_u$ between $u$ and $\alpha$ such that

$$\rho(u) = \rho(\alpha) + (u - \alpha)\rho'(\xi_u),$$

and hence

$$\begin{aligned}
\iota_h(x) &= (\psi_h \circ \rho \circ \phi_h)(x) \\
&= \psi_h\left(\rho(\alpha) + hx\rho'(\xi_{hx+\alpha})\right) \\
&= \frac{x\rho'(\xi_{hx+\alpha})}{\rho'(\alpha)}
\end{aligned}$$

for $h$ sufficiently small that $\phi_h(L) \subseteq [a,b]$.

Now let $\rho'$ have modulus of continuity $\omega$ on $[a,b]$. Let $\iota\colon \mathbb{R} \to \mathbb{R}$ represent the identity function. Then for all $x \in L$,

$$\begin{aligned}
|\iota_h(x) - \iota(x)| &= |x|\left|\frac{\rho'(\xi_{hx+\alpha}) - \rho'(\alpha)}{\rho'(\alpha)}\right| \\
&\leqslant \frac{|x|}{|\rho'(\alpha)|}\omega(hx),
\end{aligned}$$

and so $\iota_h \to \iota$ uniformly over $L$. $\qquad\square$

**Notation.** Throughout the rest of this paper $\iota_h$ will be used to denote such an approximation to the identity function, where $\iota_h \to \iota$ uniformly as $h \to 0$.

An enhnaced neuron may be described as performing (for example) the computation $x \mapsto \iota_h(4x+3)$. This is possible as the affine transformation $x \mapsto 4x + 3$ and the affine transformation $\phi_h$ (from the description of $\iota_h$) may be combined together into a single affine transformation.

## 4.2 Nonpolynomial activation functions

We consider the 'Register Model', which represents a simplification of a neural network.

**Proposition 4.3** (Register Model). *Let $\rho\colon \mathbb{R} \to \mathbb{R}$ be any continuous nonpolynomial function. Let $\mathcal{I}^\rho_{n,m,n+m+1}$ represent the class of neural networks with $n$ neurons in the input layer, $m$ neurons in the output layer, $n+m+1$ neurons in each hidden layer, an arbitrary number of hidden layers, and for which $n + m$ of the neurons in each hidden layer have the identity activation function, and one neuron in each hidden layer has activation function $\rho$. Let $K \subseteq \mathbb{R}^n$ be compact. Then $\mathcal{I}^\rho_{n,m,n+m+1}$ is dense in $C(K; \mathbb{R}^m)$.*

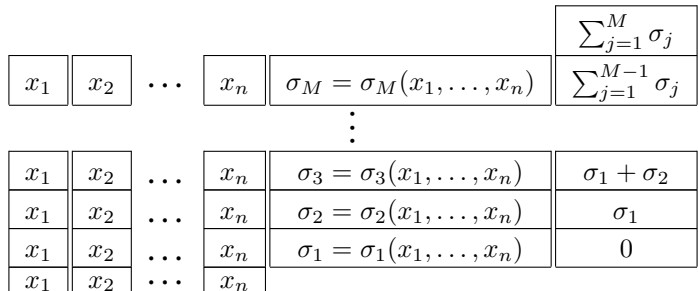

Figure 1: A simple example of how to prove the Register Model. The values $x_1, \ldots, x_n$ are inputs to the network, and the value $\sum_{j=1}^{M} \sigma_j$ is the output. Each cell represents one neuron. Each $\sigma_i$ is of the form $\psi_i \circ \rho \circ \phi_i$, where $\psi_i$ and $\phi_i$ are affine functions and $\rho$ is the activation function.

See Appendix A for the proof.

A simplified depiction of the proof of the Register Model is shown in Figure 1, for the special case of $m = 1$. It uses $n$ neurons in each layer as registers to preserve the input values. A single neuron in each layer performs a computation based off of the input values, which were preserved in the previous layer. The remaining neuron in each layer also acts a register, gradually summing up the results of the computation neurons. The computation neurons may be shown to exist by the classical Universal Approximation Theorem.

The Register Model is similar to Hanin & Sellke (2017), who have a related construction specific to the ReLU. The idea of the Register Model may also be thought of as thematically similar to residual networks, as in Lin & Jegelka (2018): in both cases the network is almost applying the identity transformation at each layer, with only a small amount of nonlinearity.

**Theorem 4.4.** *Let $\rho \colon \mathbb{R} \to \mathbb{R}$ be any continuous nonpolynomial function which is continuously differentiable at at least one point, with nonzero derivative at that point. Let $K \subseteq \mathbb{R}^n$ be compact. Then $\mathcal{NN}_{n,m,n+m+1}^{\rho}$ is dense in $C(K; \mathbb{R}^m)$.*

*Proof.* Let $f \in C(K; \mathbb{R}^m)$ and $\varepsilon > 0$. Set up a neural network as in the Register Model (Proposition 4.3), approximating $f$ to within $\varepsilon/2$. Every neuron requiring an identity activation function in the Register Model will instead approximate the identity, in the manner of Lemma 4.2.

Uniform continuity preserves uniform convergence, compactness is preserved by continuous functions, and a composition of two uniformly convergent sequences of functions with uniformly continuous limits is again uniformly convergent. So as a neural network is a layer-by-layer composition of functions then the new model can be taken within $\varepsilon/2$ of the Register Model, with respect to $\|\cdot\|_\infty$ in $K$, by taking $h$ sufficiently small. $\quad\square$

**Remark 4.5.** This of course implies approximation in $L^p(K, \mathbb{R}^m)$ for $p \in [1, \infty)$. However, when $\rho$ is the ReLU activation function, then the next corollary shows that in fact the result may be generalised to unbounded domains.

**Corollary 4.6.** *Let $\rho$ be the ReLU activation function. Let $p \in [1, \infty)$. Then $\mathcal{NN}_{n,m,n+m+1}^{\rho}$ is dense in $L^p(\mathbb{R}^n; \mathbb{R}^m)$.*

See Appendix B for the proof.

Given some $f \in L^p(\mathbb{R}^n; \mathbb{R}^m)$, the essential idea of the proof is to choose a compact set $K \subseteq \mathbb{R}^n$ on which $f$ places most of its mass, and find a neural approximation to $f$ on $K$ in the manner of Theorem 4.4. Once this is done, a cut-off function is applied outside the set, so that the network takes the value zero in $\mathbb{R}^n \setminus K$. The interesting bit is finding a neural representation of such cut-off behaviour.

In particular the 'obvious' thing to do – multiply by a cut-off function – does not appear to have a suitable neural representation, as merely approximating the multiplication operation is not necessarily enough on an unbounded domain. Instead the strategy is to take a maximum and a minimum with suitable cut-off functions.

### 4.3 POLYNOMIAL ACTIVATION FUNCTIONS

For the classical Universal Approximation Theorem, it was necessary that the activation function be nonpolynomial. However that turns out to be unnecessary here; deep narrow networks are different to shallow wide networks, and polynomial activations functions are reasonable choices.

We begin with the simplest possible nonaffine polynomial, namely $\rho(x) = x^2$.

**Proposition 4.7** (Square Model). *Let $\rho(x) = x^2$. Let $K \subseteq \mathbb{R}^n$ be compact. Then $\mathcal{NN}^{\rho}_{n,m,n+m+1}$ is dense in $C(K; \mathbb{R}^m)$.*

See Appendix C for the proof. As might be expected, density is established with the help of the Stone–Weierstrass theorem, reducing the problem to the approximation of arbitrary polynomials.

We remark that it is actually straightforward to find a construction showing that $\mathcal{NN}^{\rho}_{n,m,n+m+2}$ is dense in $C(K; \mathbb{R}^m)$ when $\rho(x) = x^2$, note the increased width. This is because the square activation function can be used to perform multiplication, via $xy = ((x+y)^2 - (x-y)^2)/4$, and this makes it easy to construct arbitrary polynomials. In fact this is what is done in the proof of Proposition 4.7 for finding $m - 1$ of the $m$ outputs, when there is still a 'spare' neuron in each layer. It is computing the final output that actually requires the bulk of the work. The key to this argument is a width-efficient approximation to division.

It is a consequence of Proposition 4.7 that any (polynomial) activation function which can approximate the square activation function, in a suitable manner, is also capable of universal approximation.

**Theorem 4.8.** *Let $\rho \colon \mathbb{R} \to \mathbb{R}$ be any polynomial for which there exists a point $\alpha \in \mathbb{R}$ such that $\rho'(\alpha) = 0$ and $\rho''(\alpha) \neq 0$. Let $K \subseteq \mathbb{R}^n$ be compact. Then $\mathcal{NN}^{\rho}_{n,m,n+m+1}$ is dense in $C(K; \mathbb{R}^m)$.*

*Proof.* Let $h \in \mathbb{R} \setminus \{0\}$. Define $\rho_h \colon \mathbb{R} \to \mathbb{R}$ by

$$\rho_h(x) = \frac{\rho(\alpha + hx) - \rho(\alpha)}{h^2 \rho''(\alpha)/2}.$$

Then, taking a Taylor explansion around $\alpha$,

$$\rho_h(x) = \frac{\rho(\alpha) + hx\rho'(\alpha) + h^2 x^2 \rho''(\alpha)/2 + \mathcal{O}(h^3 x^3) - \rho(\alpha)}{h^2 \rho''(\alpha)/2}$$

$$= x^2 + \mathcal{O}(hx^3).$$

Let $s(x) = x^2$. Then $\rho_h \to s$ uniformly over any compact set as $h \to 0$.

Now set up a network as in the Square Model (Proposition 4.7), with every neuron using the square activation function. Call this network $N$. Create a network $N_h$ by copying $N$ and giving every neuron in the network the activation function $\rho_h$ instead.

Uniform continuity preserves uniform convergence, compactness is preserved by continuous functions, and a composition of two uniformly convergent sequences of functions with uniformly continuous limits is again uniformly convergent. So as a neural network is a layer-by-layer composition of functions, then the difference between $N$ and $N_h$, with respect to $\| \cdot \|_\infty$ on $K$, may be taken arbitrarily small by taking $h$ arbitrarily small.

Furthermore note that $\rho_h$ is just $\rho$ pre- and post-composed with affine functions. (Note that there is only one term in the definition of $\rho_h(x)$ which depends on $x$.) This means that any network which may be represented with activation function $\rho_h$ may be precisely represented with activation function $\rho$, by combining the affine transformations involved. $\square$

**Remark 4.9.** That $\rho$ is polynomial is never really used in the proof of Theorem 4.8. Only a certain amount of differentiability is required, and all such nonpolynomial functions are already covered by Theorem 4.4, as a nonzero second derivative at $\alpha$ implies a nonzero first derivative somewhere close to $\alpha$. Nonetheless in principle this provides another possible construction by which certain networks may be shown to exhibit universal approximation.

Note that the converse strategy (applying nonpolynomial techniques to the polynomial case) fails. This is because the Register Model requires nonpolynomial activation functions due to its dependence on the classical Universal Approximation Theorem.

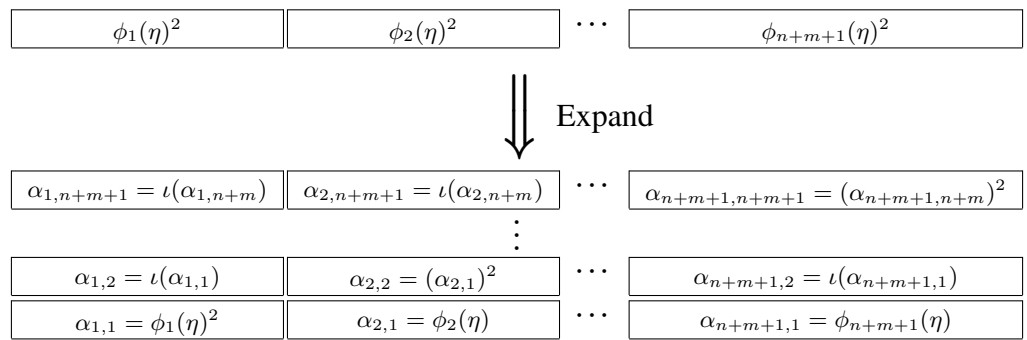

Figure 2: A layer with square activation functions is equivalent to multiple layers with only a single square activation function in each layer. The other neurons use the identity activation function, denoted $\iota$.

**Theorem 4.10.** *Let $\rho\colon \mathbb{R} \to \mathbb{R}$ be any nonaffine polynomial. Let $K \subseteq \mathbb{R}^n$ be compact. Then $\mathcal{NN}^{\rho}_{n,m,n+m+2}$ is dense in $C(K;\mathbb{R}^m)$.*

*Proof.* Fix $\alpha \in \mathbb{R}$ such that $\rho''(\alpha) \neq 0$, which exists as $\rho$ is nonaffine. Now let $h \in (0,\infty)$. Define $\sigma_h\colon \mathbb{R} \to \mathbb{R}$ by

$$\sigma_h(x) = \frac{\rho(\alpha + hx) - 2\rho(\alpha) + \rho(\alpha - hx)}{h^2 \rho''(\alpha)}.$$

Then Taylor expanding $\rho(\alpha + hx)$ and $\rho(\alpha - hx)$ around $\alpha$,

$$\sigma_h(x) = \frac{\rho(\alpha) + hx\rho'(\alpha) + h^2 x^2 \rho''(\alpha)/2 + \mathcal{O}(h^3 x^3)}{h^2 \rho''(\alpha)} - \frac{2\rho(\alpha)}{h^2 \rho''(\alpha)} + $$
$$\frac{\rho(\alpha) - hx\rho'(\alpha) + h^2 x^2 \rho''(\alpha)/2 + \mathcal{O}(h^3 x^3)}{h^2 \rho''(\alpha)}$$
$$= x^2 + \mathcal{O}(hx^3).$$

Observe that $\sigma_h$ needs precisely two operations of $\rho$ on (affine transformations of) $x$, and so may be computed by two enhanced neurons with activation function $\rho$. Thus the operation of a single enhanced neuron with square activation function may be approximated by two enhanced neurons with activation function $\rho$.

Let $N$ be a network as in the Square Model (Proposition 4.7) with every neuron using the square activation function. Let $\ell$ be any hidden layer of $N$; it contains $n+m+1$ neurons. Let $\eta$ be a vector of the values of the neurons of the previous layer. Let $\phi_i$ be the affine part of the $i$th neuron of $\ell$, so that $\ell$ computes $\phi_1(\eta)^2, \ldots, \phi_{n+m+1}(\eta)^2$. Then this may equivalently be calculated with $n+m+1$ layers of $n+m+1$ neurons each, with $n+m$ of the neurons in each of these new layers using the identity function, and one neuron using the square activation function. The first of these new layers applies the $\phi_i$, and the $i$th layer squares the value of the $i$th neuron. See Figure 2.

Apply this procedure to every layer of $N$; call the resulting network $\widetilde{N}$. It will compute exactly the same function as $N$, and will have $n+m+1$ times as many layers, but will use only a single squaring operation in each layer.

Create a copy of $\widetilde{N}$, call it $\widetilde{N}_h$. Replace its identity activation functions with approximations in the manner of Lemma 4.2, using activation function $\rho$. Replace its square activation functions (one in each layer) by approximations in the manner described above with $\sigma_h$; this requires an extra neuron in each hidden layer, so that the network is now of width $n+m+2$. Thus $\widetilde{N}_h$ uses the activation function $\rho$ throughout.

Uniform continuity preserves uniform convergence, compactness is preserved by continuous functions, and a composition of two uniformly convergent sequences of functions with uniformly continuous limits is again uniformly convergent. So as a neural network is a layer-by-layer composition

of functions, then the difference between $\widetilde{N}_h$ and $\widetilde{N}$, with respect to $\|\cdot\|_\infty$ on $K$, may be taken arbitrarily small by taking $h$ arbitrarily small. $\qquad\square$

**Remark 4.11.** It is possible to construct shallower networks analogous to $\widetilde{N}$. The proof of Proposition 4.7 in Appendix C, uses most of the network's neurons to approximate the identity anyway; only a few in each layer are used to square a valued that is desired to be squared. These are the only neurons that actually require the procedure used in Figure 2 and the proof of Theorem 4.10.

## 4.4 NONDIFFERENTIABLE ACTIVATION FUNCTIONS

Although not of direct practical application, results for nondifferentiable activation functions demonstrate how certain pathological cases are still capable of being handled.

**Lemma 4.12.** *Let $w\colon \mathbb{R} \to \mathbb{R}$ be any bounded continuous nowhere differentiable function. Let $\rho(x) = \sin(x) + w(x)\mathrm{e}^{-x}$. Let $L \subseteq \mathbb{R}$ be compact. Then a single enhanced neuron with activation function $\rho$ may uniformly approximate the identity function $\iota\colon \mathbb{R} \to \mathbb{R}$ on L, with arbitrarily small error.*

*Proof.* For $h \in \mathbb{R} \setminus \{0\}$ and $A \in 2\pi\mathbb{N}$, let $\phi_{h,A}(x) = hx + A$, and let $\psi(x) = x/h$. Let

$$\iota_{h,A} = \psi_h \circ \rho \circ \phi_{h,A},$$

which is of the form that an enhanced neuron can represent. Then jointly taking $h$ small enough and $A$ large enough it is clear that $\iota_{h,A}$ may be taken uniformly close to $\iota$ on $L$. $\qquad\square$

**Proposition 4.13.** *Let $w\colon \mathbb{R} \to \mathbb{R}$ be any bounded continuous nowhere differentiable function. Let $\rho(x) = \sin(x) + w(x)\mathrm{e}^{-x}$, which will also be nowhere differentiable. Let $K \subseteq \mathbb{R}^n$ be compact. Then $\mathcal{NN}^{\rho}_{n,m,n+m+1}$ is dense in $C(K;\mathbb{R}^m)$.*

*Proof.* As the proof of Theorem 4.4, except substituting Lemma 4.12 for Lemma 4.2. $\qquad\square$

This manner of proof may be extended to other nondifferentiable activation functions as well.

## 5 CONCLUSION

There is a large literature on theoretical properties of neural networks, but much of it deals only with the ReLU.[1] However how to select an activation function remains a poorly understood topic, and many other options have been proposed: leaky ReLU, PReLU, RRelu, ELU, SELU and other more exotic activation functions as well.[2]

Our central contribution is to provide results for universal approximation using general activation functions (Theorems 3.2, 4.4, 4.8 and 4.10). In contrast to previous work, these results do not rely on the nice properties of the ReLU, and in particular do not rely on its explicit description. The techniques we use are straightforward, and robust enough to handle even the pathological case of nondifferentiable activation functions (Proposition 4.13).

We also consider approximation in $L^p$ norm (Remark 4.5), and generalise previous work to smaller widths, multiple output neurons, and $p \geqslant 1$ in place of $p = 1$ (Corollary 4.6).

In contrast to much previous work, every result we show also handles the general case of multiple output neurons.

ACKNOWLEDGEMENTS

(Redacted from anonymised submission)

---

[1]See for example Hanin & Sellke (2017); Petersen & Voigtlaender (2018); Gühring et al.; Daubechies et al. (2019); Arora et al. (2018).

[2]See Maas et al. (2013); He et al. (2015); Xu et al. (2015); Clevert et al. (2016); Klambauer et al. (2017); Molina et al. (2019); Krizhevsky (2012) respectively.

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

## A   PROOF OF THE REGISTER MODEL (PROPOSITION 4.3)

First, we recall the classical Universal Approximation Theorem (Pinkus, 1999):

**Theorem 1.1.** *Let $\rho\colon \mathbb{R} \to \mathbb{R}$ be any continuous function. Let $\mathcal{N}_n^\rho$ represent the class of neural networks with activation function $\rho$, with $n$ neurons in the input layer, one neuron in the output layer, and one hidden layer with an arbitrary number of neurons. Let $K \subseteq \mathbb{R}^n$ be compact. Then $\mathcal{N}_n^\rho$ is dense in $C(K)$ if and only if $\rho$ is nonpolynomial.*

The Register Model is created by suitably reorganising the neurons from a collection of such shallow networks.

**Proposition 4.3** (Register Model)**.** *Let $\rho\colon \mathbb{R} \to \mathbb{R}$ be any continuous nonpolynomial function. Let $\mathcal{I}_{n,m,n+m+1}^\rho$ represent the class of neural networks with $n$ neurons in the input layer, $m$ neurons in the output layer, $n+m+1$ neurons in each hidden layer, an arbitrary number of hidden layers, and for which $n+m$ of the neurons in each hidden layer have the identity activation function, and one neuron in each hidden layer has activation function $\rho$. Let $K \subseteq \mathbb{R}^n$ be compact. Then $\mathcal{I}_{n,m,n+m+1}^\rho$ is dense in $C(K; \mathbb{R}^m)$.*

*Proof.* Fix $f \in C(K; \mathbb{R}^m)$. Let $f = (f_1, \ldots, f_m)$. Fix $\varepsilon > 0$. By Theorem 1.1, there exist single-hidden-layer neural networks $g_1, \ldots, g_m \in \mathcal{N}_n^\rho$ with activation function $\rho$ approximating $f_1, \ldots, f_m$ respectively. Each approximation is to within error $\varepsilon$ with respect to $\|\cdot\|_\infty$ on $K$. Let each $g_i$ have $\beta_i$ hidden neurons. Let $\sigma_{i,j}$ represent the operation of its $j$th hidden neuron, for $j \in \{1, \ldots, \beta_i\}$. In keeping with the idea of enhanced neurons, let each $\sigma_{i,j}$ include the affine function that comes after it in the output layer of $g_i$, so that $g_i = \sum_{j=1}^{\beta_i} \sigma_{i,j}$. Let $M = \sum_{i=1}^m \beta_i$.

We seek to construct a neural network $N \in \mathcal{I}_{n,m,n+m+1}^\rho$. Given input $(x_1, \ldots, x_n) \in \mathbb{R}^n$, it will output $(G_1, \ldots, G_m) \in \mathbb{R}^m$, such that $G_i = g_i(x_1, \ldots, x_n)$ for each $i$. That is, it will compute all of the shallow networks $g_1, \ldots, g_m$. Thus it will approximate $f$ to within error $\varepsilon$ with respect to $\|\cdot\|_\infty$ on $K$.

The construction of $N$ is mostly easily expressed pictorially; see Figure 3. In each cell, representing a neuron, we define its value as a function of the values of the neurons in the previous layer. In every layer, all but one of the neurons uses the identity activation function $\iota\colon \mathbb{R} \to \mathbb{R}$, whilst one neuron in each layer performs a computation of the form $\sigma_{i,j}$.

The construction can be summed up as follows.

Each layer has $n + m + 1$ neurons, arranged into a group of $n$ neurons, a group of a single neuron, and a group of $m$ neurons.

The first $n$ neurons in each layer simply record the input $(x_1, \ldots, x_n)$, by applying an identity activation function. We refer to these as the 'in-register neurons'.

Next we consider $g_1, \ldots, g_m$, which are all shallow networks. The neurons in the hidden layers of $g_1, \ldots, g_m$ are arranged 'vertically' in our deep network, one in each layer. This is the neuron in each layer that uses the activation function $\rho$. We refer to these as the 'computation neurons'. Each computation neuron performs its computation based off of the inputs preserved in the in-register neurons.

The final group of $m$ neurons also use the identity activation function; their affine parts gradually sum up the results of the computation neurons. We refer to these as the 'out-register neurons'. The $i$th out-register neuron in each layer will sum up the results of the computation neurons computing $\sigma_{i,j}$ for all $j \in \{1, \ldots, \beta_i\}$.

Finally, the neurons in the output layer of the network are connected to the out-register neurons of the final hidden layer. As each of the neurons in the output layer has, as usual, the identity activation function, they will now have computed the desired results. □

Figure 3: The thick lines delimit groups of layers; the $i$th group computes $\sigma_{i,1}, \ldots, \sigma_{i,\beta_i}$. The inputs to the network are $x_1, \ldots, x_n$, depicted at the bottom. The outputs from the network are $G_1, \ldots, G_m$, depicted at the top. The identity activation function $\mathbb{R} \to \mathbb{R}$ is denoted $\iota$.

# B  PROOF OF COROLLARY 4.6

**Lemma B.1.** *Let $a, b, c, d \in \mathbb{R}$ be such that $a < b < c < d$. Let $U_{a,b,c,d} \colon \mathbb{R} \to \mathbb{R}$ be the unique continuous piecewise affine function which is one on $[b, c]$ and zero on $(-\infty, a] \cup [d, \infty)$. Then two layers of two enhanced neurons each, with ReLU activation function, may exactly represent the function $U_{a,b,c,d}$.*

*Proof.* Let $x \in \mathbb{R}$ be the input. Let $m_1 = 1/(b - a)$. Let $m_2 = 1/(d - c)$. Let $\eta_1, \eta_2$ represent the first neuron in each layer, and $\zeta_1, \zeta_2$ represent the second neuron in each layer. We assign them values as follows.

$$\eta_1 = \max\{0, m_1(x - a)\}, \qquad\qquad \zeta_1 = \max\{0, m_2(x - c)\},$$
$$\eta_2 = \max\{0, 1 - \eta_1\}, \qquad\qquad \zeta_2 = \max\{0, 1 - \zeta_1\}.$$

Then $U_{a,b,c,d}(x) = \zeta_2 - \eta_2$. (This final affine transformation is allowed, in keeping with the notion of enhanced neurons.) $\qquad\square$

**Lemma B.2.** *One layer of two enhanced neurons, with ReLU activation function, may exactly represent the function $(x, y) \mapsto \min\{x, y\}$ on $[0, \infty)^2$.*

*Proof.* Let the first neuron compute $\eta = \max\{0, x - y\}$. Let the second neuron compute $\zeta = \max\{0, x\}$. Then $\min\{x, y\} = \zeta - \eta$. $\qquad\square$

**Corollary 4.6.** *Let $\rho$ be the ReLU activation function. Let $p \in [1, \infty)$. Then $\mathcal{NN}^\rho_{n,m,n+m+1}$ is dense in $L^p(\mathbb{R}^n; \mathbb{R}^m)$.*

*Proof.* Let $f \in L^p(\mathbb{R}^n; \mathbb{R}^m)$ and $\varepsilon > 0$. For simplicity assume that $\mathbb{R}^m$ is endowed with the $\|\cdot\|_\infty$ norm; other norms are of course equivalent. Let $\widehat{f} = (\widehat{f}_1, \ldots, \widehat{f}_m) \in C_c(\mathbb{R}^n; \mathbb{R}^m)$ be such that

$$\left\| f - \widehat{f} \right\|_p < \varepsilon/3. \tag{B.1}$$

Let

$$C = \sup_{x \in \mathbb{R}^n} \max_i \widehat{f}_i(x) + 1 \tag{B.2}$$

and

$$c = \inf_{x \in \mathbb{R}^n} \min_i \widehat{f}_i(x) - 1 \tag{B.3}$$

Pick $a_1, b_1, \ldots, a_n, b_n \in \mathbb{R}$ such that $J$ defined by

$$J = [a_1, b_1] \times \cdots \times [a_n, b_n]$$

is such that $\operatorname{supp} \widehat{f} \subseteq J$. Furthermore, for $\delta > 0$ that we shall fix in a moment, let

$$A_i = a_i - \delta,$$
$$B_i = b_i + \delta,$$

and let $K$ be defined by

$$K = [A_1, B_1] \times \cdots \times [A_n, B_n].$$

Fix $\delta$ small enough that

$$|K \setminus J|^{1/p} \cdot \max\{|C|, |c|\} < \frac{\varepsilon}{6}. \tag{B.4}$$

Let $g = (g_1, \ldots, g_m) \in \mathcal{NN}^\rho_{n,m,n+m+1}$ be such that

$$\sup_{x \in K} \left| \widehat{f}(x) - g(x) \right| < \min\left\{ \frac{\varepsilon}{3|J|^{1/p}}, 1 \right\}, \tag{B.5}$$

which exists by Theorem 4.4. Note that $g$ is defined on all of $\mathbb{R}^n$; it simply happens to be close to $\widehat{f}$ on $K$. In particular it will takes values close to zero on $K \setminus J$, and may take arbitrary values in $\mathbb{R}^n \setminus K$. By equations (B.2), (B.3), (B.5), it is the case that

$$C \geqslant \sup_{x \in K} \max_i g_i(x),$$
$$c \leqslant \inf_{x \in K} \min_i g_i(x). \tag{B.6}$$

Now consider the network describing $g$; it will be modified slightly. The goal is to create a network which takes value $g$ on $J$, zero in $\mathbb{R}^n \setminus K$, and moves between these values in the interface region $K \setminus J$. Such a network will provide a suitable approximation to $\widehat{f}$. This is done by first constructing a function which is approximately the indicator function for $J$, with support in $K$; call such a function $U$. The idea then is to construct a neural representation of $G_i$ defined by

$$G_i = \min\{\max\{g_i, cU\}, CU\}.$$

Provided $|K \setminus J|$ is small enough then $G = (G_1, \ldots, G_m)$ will be the desired approximation; this is proved this below.

We move on to presenting the neural representation of this construction.

First we observe that because the activation function is the ReLU, then the identity approximations used in the proof of Theorem 4.4 may in fact exactly represent the identity function on some compact set: $x \mapsto \max\{0, x + N\} - N$ is exactly the identity function, for suitably large $N$, and is of the form that an enhanced neuron may represent. This observation isn't strictly necessary for the proof, but it does simplify the presentation somewhat, as the values preserved in the in-register neurons of $g$ are now exactly the inputs $x = (x_1, \ldots, x_n)$ for $x \in K$. For sufficiently negative $x_i$, outside of $K$, they will take the value $-N$ instead, but by insisting that is $N$ sufficiently large that

$$-N < A_i \tag{B.7}$$

for all $i$, then this will not be an issue for the proof.

So take the network representing $g$, and remove the output layer. (If the output layer is performing any affine transformations then treat them as being part of the final hidden layer, in the manner of enhanced neurons. Thus the output layer that is being removed is just applying the identity function to the out-register neurons.) Some more hidden layers will be placed on top, and then a new output layer will be placed on top. In the following description, all neurons not otherwise specified will be performing the identity function, so as to preserve the values of the corresponding neurons in the preceding layer. As all functions involved are continuous and $K$ is compact, and compactness is preserved by continuous functions, and continuous functions are bounded on compact sets, then this is possible for all $x \in K$ by taking $N$ large enough.

The first task is to modify the value stored in the in-register neurons corresponding to $x_1$. At present it stores the value $x_1$; by using this in-register neuron and the computation neuron in two extra layers, its value may be replaced with $U_{A_1, a_1, b_1, B_1}(x_1)$, via Lemma B.1. Place another two layers on top, and use them to replace the value of $x_2$ in the second in-register neuron with $U_{A_2, a_2, b_2, B_2}(x_2)$, and so on. The in-register neurons now store the values $(U_{A_1, a_1, b_1, B_1}(x_1), \ldots, U_{A_n, a_n, b_n, B_n}(x_n))$.

Once this is complete, place another layer on top and use the first two in-register neurons to compute the minimum of their values, in the manner of Lemma B.2, thus computing $\min\{U_{A_1, a_1, b_1, B_1}(x_1), U_{A_2, a_2, b_2, B_2}(x_2)\}$. Place another layer on top and use another two in-register neurons to compute the minimum of this value and the value presently stored in the third in-register neuron, that is $U_{A_3, a_3, b_3, B_3}(x_3)$, so that

$$\min\{U_{A_1, a_1, b_1, B_1}(x_1), U_{A_2, a_2, b_2, B_2}(x_2), U_{A_3, a_3, b_3, B_3}(x_3)\}$$

has now been computed. Continue to repeat this process until the in-register neurons have computed.[3]

$$U = \min_{i \in \{1, \ldots, n\}} U_{A_i, a_i, b_i, B_i}(x_i).$$

---

[3] It doesn't matter which of the in-register neurons records the value of $U$.

Observe how $U$ represents an approximation to the indicator function for $J$, with support in $K$, evaluated at $(x_1, \ldots, x_n)$.

This is a highly destructive set of operations: the network no longer remembers the values of its inputs. Thankfully, it no longer needs them. Note how the small foible regarding how an in-register neuron would only record $-N$ instead of $x_i$, for $x_i < -N$, is not an issue. This is because of equation (B.7), which implies that $U_{A_i, a_i, b_i, B_i}(x_i) = 0 = U_{A_i, a_i, b_i, B_i}(-N)$, thus leaving the value of $U$ unaffected.

The out-register neurons presently store the values $g_1, \ldots, g_m$, where $g_i = g_i(x_1, \ldots, x_n)$. Now add another layer. Let the value of its out-register neurons be $\theta_1, \ldots, \theta_m$, where

$$\theta_i = \max\{0, g_i - cU\}.$$

Add one more hidden layer. Let the value of its out-register neurons be $\lambda_1, \ldots, \lambda_m$, where

$$\lambda_i = \max\{0, -\theta_i + (C - c)U\}.$$

Finally place the output layer on top. Let the value of its neurons be $G_1, \ldots, G_m$, where

$$G_i = -\lambda_i + CU.$$

Then in fact

$$G_i = \min\{\max\{g_i, cU\}, CU\} \tag{B.8}$$

as desired.

All that remains to show is that $G = (G_1, \ldots, G_m)$ of this form is indeed a suitable approximation. First, as $G$ and $g$ coincide in $J$, and by equation (B.5),

$$\left( \int_J \left| \widehat{f}(x) - G(x) \right|^p dx \right)^{1/p} \leqslant |J|^{1/p} \sup_{x \in J} \left| \widehat{f}(x) - G(x) \right|$$

$$= |J|^{1/p} \sup_{x \in J} \left| \widehat{f}(x) - g(x) \right|$$

$$< \frac{\varepsilon}{3}. \tag{B.9}$$

Secondly, by equations (B.2), (B.3), (B.6), (B.8) and then equation (B.4),

$$\left( \int_{K \setminus J} \left| \widehat{f}(x) - G(x) \right|^p dx \right)^{1/p} \leqslant |K \setminus J|^{1/p} \sup_{x \in K \setminus J} \left| \widehat{f}(x) - G(x) \right|$$

$$\leqslant |K \setminus J|^{1/p} \cdot 2 \max\{|C|, |c|\}$$

$$< \frac{\varepsilon}{3}. \tag{B.10}$$

Thirdly,

$$\left( \int_{\mathbb{R}^n \setminus K} \left| \widehat{f}(x) - G(x) \right|^p dx \right)^{1/p} = 0, \tag{B.11}$$

as both $\widehat{f}$ and $G$ have support in $K$.

So by equations (B.1), (B.9), (B.10) and (B.11),

$$\|f - G\|_p \leqslant \left\| f - \widehat{f} \right\|_p + \left( \int_{\mathbb{R}^n} \left| \widehat{f}(x) - G(x) \right|^p dx \right)^{1/p}$$

$$< \frac{\varepsilon}{3} + \frac{\varepsilon}{3} + \frac{\varepsilon}{3}$$

$$= \varepsilon. \qquad \square$$

## C  PROOF OF THE SQUARE MODEL (PROPOSITION 4.7)

**Lemma C.1.** *One layer of two enhanced neurons, with square activation function, may exactly represent the multiplication function* $(x, y) \mapsto xy$ *on* $\mathbb{R}^2$.

*Proof.* Let the first neuron compute $\eta = (x+y)^2/4$. Let the second neuron compute $\zeta = (x-y)^2/4$. Then $xy = \eta - \zeta$. $\qquad\square$

**Lemma C.2.** *Fix* $L \subseteq \mathbb{R}^2$ *compact. Three layers of two enhanced neurons each, with square activation function, may uniformly approximate* $(x, y) \mapsto (x^2, y(x+1))$ *arbitrarily well on* $L$.

*Proof.* Let $h, s \in \mathbb{R} \setminus \{0\}$. Let $\eta_1, \eta_2, \eta_3$ represent the first neuron in each layer; let $\zeta_1, \zeta_2, \zeta_3$ represent the second neuron in each layer. Let $\iota_h$ represent an approximation to the identity in the manner of Lemma 4.2. Using '$\approx$' as an informal notation to represent 'equal to up to the use of $\iota_h$ in place $\iota$', just to help keep track of why we are performing these operations, assign values to $\eta_1, \eta_2, \eta_3$ and $\zeta_1, \zeta_2, \zeta_3$ as follows:

$$
\begin{aligned}
\eta_1 &= \iota_h(x) & \zeta_1 &= (x + sy + 1)^2 \\
&\approx x, & &= x^2 + 2sxy + s^2y^2 + 2x + 2sy + 1, \\
\eta_2 &= (\eta_1)^2 & \zeta_2 &= \iota_h(\zeta_1 - 2\eta_1 - 1) \\
&\approx x^2, & &\approx x^2 + 2sxy + s^2y^2 + 2sy, \\
\eta_3 &= \iota_h(\eta_2) & \zeta_3 &= \iota_h((\zeta_2 - \eta_2)/2s) \\
&\approx x^2, & &\approx xy + y + sy^2/2.
\end{aligned}
$$

And so $\eta_3$ may be taken arbitrarily close to $x^2$ and $\zeta_3$ may be taken arbitrarily close to $y(x+1)$, with respect to $\|\cdot\|_\infty$ on $L$, by first taking $s$ arbitrarily small, and then taking $h$ arbitrarily small. $\qquad\square$

**Proposition C.3.** *Fix* $L \subseteq (0, 2)$ *compact. Then multiple layers of two enhanced neurons each, with square activation function, may uniformly approximate* $x \mapsto 1/x$ *arbitrarily well on* $L$.

(Unlike Lemma C.2, the number of layers necessary will depend on the quality of approximation.)

*Proof.* First note that

$$
\prod_{i=0}^{n}(1 + x^{2^i}) \to \frac{1}{1-x}
$$

as $n \to \infty$, uniformly over compact subsets of $(-1, 1)$. Thus,

$$
(2 - x)\prod_{i=1}^{n}(1 + (1 - x)^{2^i}) = \prod_{i=0}^{n}(1 + (1 - x)^{2^i}) \to \frac{1}{x}
$$

uniformly over $L$.

This has the following neural approximation: let $\eta_1 = (1-x)^2$ and $\zeta_1 = \iota_h(2-x)$ be the neurons in the first layer, where $\iota_h$ is some approximation of the identity as in Lemma 4.2. Let $\kappa_h$ represent an approximation to $(x, y) \mapsto (x^2, y(x+1))$ in the manner of Lemma C.2, with error made arbitrarily small as $h \to 0$. Now for $i \in \{1, 4, 7, 10, \dots, 3n - 2\}$, recursively define $(\eta_{i+3}, \zeta_{i+3}) = \kappa_h(\eta_i, \zeta_i)$, where we increase the index by three to represent the fact that three layers are used to perform this operation. So up to approximation, $\eta_{i+3} \approx (\eta_i)^2$, and $\zeta_{i+3} \approx \zeta_i(\eta_i + 1)$.

So $\zeta_{3n+1} \to (2 - x)\prod_{i=1}^{n}(1 + (1 - x)^{2^i})$ uniformly over $L$ as $h \to 0$. Thus the result is obtained by taking first $n$ large enough and then $h$ small enough. $\qquad\square$

**Remark C.4.** Our approach to Proposition C.3 is to find a suitable polynomial approximation of the reciprocal function, and then represent that with a network of multiple layers of two neurons each. It is fortunate, then, that this polynomial happens to be of a form that may be represented by such a network, as it is not clear that this should necessarily be the case for all polynomials. Even

if Proposition 4.7 were already known, it requires a network of width three to represent arbitrary polynomials $\mathbb{R} \to \mathbb{R}$, whereas Proposition C.3 uses a network of only width two. It remains unclear whether an arbitrary-depth network of width two, with square activation function, is capable of universal approximation in $C(K)$.

**Proposition 4.7** (Square Model). *Let $\rho(x) = x^2$. Let $K \subseteq \mathbb{R}^n$ be compact. Then $\mathcal{NN}^{\rho}_{n,m,n+m+1}$ is dense in $C(K; \mathbb{R}^m)$.*

*Proof.* Fix $f \in C(K; \mathbb{R}^m)$. Let $f = (f_1, \ldots, f_m)$. Fix $\varepsilon > 0$. By precomposing with an affine function – which may be absorbed into the first layer of the network – assume without loss of generality that

$$K \subseteq (1, 2)^n. \tag{C.1}$$

By the Stone–Weierstrass theorem there exist polynomials $g_1, \ldots, g_m$ in $x_1, \ldots x_n$ approximating $f_1, \ldots, f_m$ to within $\varepsilon/3$ with respect to $\| \cdot \|_\infty$.

We will construct a network that evaluates arbitrarily good approximations to $g_1, \ldots, g_n$. There are a total of $n + m + 1$ neurons in each layer; group them as in the proof of the Register Model (Proposition 4.3), see Appendix A, so that in each layer there is a group of $n$ neurons that we refer to as 'in-register neurons', a single neuron that we refer to as the 'computation neuron', and a group of $m$ neurons that we refer to as the 'out-register neurons'.

As before, every in-register neuron will simply apply an approximate identity function to the corresponding in-register neuron in the previous layer, so that they preserve the inputs to the network, up to an arbitrarily good approximation of the identity. (For now, at least – later, when constructing the approximation to $g_1$, which will be the final approximation that is handled, then these neurons will be repurposed to perform that computation.) For the sake of sanity of notation, we shall suppress this detail in our notation, and refer to our neurons in later layers as having e.g. '$x_1$' as an input to them to them; in practice this means some arbitrarily good approximation to $x_1$. The out-register neurons will eventually store the desired outputs of the network; thus there is an out-register neuron in each layer 'corresponding' to each of the $g_i$.

Now suppose $m > 1$; if $m = 1$ then this paragraph and the next three paragraphs may simply be skipped. It is easy to build a network approximating $g_2, \ldots g_m$, as there is at least one 'extra' neuron per layer that is available to use: the out-register neuron corresponding to $g_1$. The strategy is as follows. Let $g_2 = \sum_{j=1}^{N} \delta_j$, where each $\delta_j$ is a monomial. Using the computation neuron and the 'extra' out-register neuron in each layer, perform successive multiplications in the manner of Lemma C.1 to compute the value of $\delta_1$. For example, if $\delta_1 = x_1^2 x_2 x_3$, then this chain of multiplications is $x_1(x_1(x_2 x_3))$. The computation neuron and the 'extra' out-register neuron in the first layer compute the multiplication $\alpha = x_2 x_3$, these neurons in the second layer compute the multiplication $\beta = x_1 \alpha$, and these neurons in the third layer compute $x_1 \beta$. This value is then stored in the out-register neuron corresponding to $g_2$ – by using the affine part of the operation of this neuron – and kept through the layers via approximate identity functions, as per Lemma 4.2.

This process is then repeated for $\delta_2$. The result is then added on – via the affine part of the operation of the out-register neuron corresponding to $g_2$ – to the value stored in the out-register neuron corresponding to $g_2$. Repeat for all $j$ until all of the $\delta_j$ have been computed and the out-register neuron stores an approximation to $g_2$. This is only an approximation in that it requires the use of approximate identity functions; other than that it is exact. As such, by taking sufficiently good approximations of the identity function, this will be a uniform approximation to $g_2$ over $K$.

Now repeat this whole process for $g_3, \ldots g_m$.

For the rest of the layers of the network, the out-register neurons corresponding to $g_2, \ldots, g_m$ will now simply apply approximate identity functions to maintain their values: these will eventually form the outputs of the network. Let these computed values be denoted $\widehat{g}_2, \ldots, \widehat{g}_m$. (With the 'hat' notation becuse of the fact that these are not the values $g_2, \ldots, g_m$, due to the approximate identity functions in between.)

The difficult bit is computing an approximation to $g_1$, as it must be done without the 'extra' neuron in each layer. In total, then, in each layer, there are $n + 2$ neurons available: the $n$ in-register neurons (which have so far been storing the inputs $x_1, \ldots x_n$), the computation neuron, and the out-register neuron corresponding to $g_1$.

Written in terms of monomials, let $g_1 = \sum_{j=1}^M \gamma_j$. Then $g_1$ may be written as

$$g_1 = \gamma_1 \left( 1 + \frac{\gamma_2}{\gamma_1} \left( 1 + \frac{\gamma_3}{\gamma_2} \left( \cdots \left( 1 + \frac{\gamma_{M-1}}{\gamma_{M-2}} \left( 1 + \frac{\gamma_M}{\gamma_{M-1}} \right) \right) \cdots \right) \right) \right).$$

Note that this description is defined over $K$, as $K$ is bounded away from the origin.

In particular, let $\gamma_j = \prod_{k=1}^n x_k^{\theta_{j,k}}$, for $\theta_{j,k} \in \mathbb{N}_0$. Substituting this in,

$$g_1 = \left[ \prod_{k=1}^n x_k^{\theta_{1,k}} \right] \left( 1 + \frac{\prod_{k=1}^n x_k^{\theta_{2,k}}}{\prod_{k=1}^n x_k^{\theta_{1,k}}} \left( 1 + \frac{\prod_{k=1}^n x_k^{\theta_{3,k}}}{\prod_{k=1}^n x_k^{\theta_{2,k}}} \left( \cdots \right. \right. \right.$$

$$\left. \left. \left. \left( 1 + \frac{\prod_{k=1}^n x_k^{\theta_{M-1,k}}}{\prod_{k=1}^n x_k^{\theta_{M-2,k}}} \left( 1 + \frac{\prod_{k=1}^n x_k^{\theta_{M,k}}}{\prod_{k=1}^n x_k^{\theta_{M-1,k}}} \right) \right) \cdots \right) \right) \right).$$

Now let $\sup K$ be defined by

$$\sup K = \sup\{ x_i \mid (x_1, \ldots, x_n) \in K \},$$

so that $1 < \sup K < 2$. Let $r$ be an approximation to $x \mapsto 1/x$ in the manner of Proposition C.3, with the $L$ of that proposition given by

$$L = [(\sup K)^{-1} - \alpha, \sup K + \alpha] \subseteq (0, 2), \tag{C.2}$$

where $\alpha > 0$ is taken small enough that the inclusion holds.

Let $r^a$ denote $r$ composed $a$ times. By taking $r$ to be a suitably good approximation, we may ensure that $\widetilde{g}_1$ defined by

$$\widetilde{g}_1 = \left[ \prod_{k=1}^n r^{2M-2}(x_k)^{\theta_{1,k}} \right] \left( 1 + \left[ \prod_{k=1}^n r^{2M-3}(x_k)^{\theta_{1,k}} \right] \left[ \prod_{k=1}^n r^{2M-4}(x_k)^{\theta_{2,k}} \right] \right.$$

$$\left( 1 + \left[ \prod_{k=1}^n r^{2M-5}(x_k)^{\theta_{2,k}} \right] \left[ \prod_{k=1}^n r^{2M-6}(x_k)^{\theta_{3,k}} \right] \right.$$

$$\left( \cdots \right.$$

$$\left( 1 + \left[ \prod_{k=1}^n r^3(x_k)^{\theta_{M-2,k}} \right] \left[ \prod_{k=1}^n r^2(x_k)^{\theta_{M-1,k}} \right] \right.$$

$$\left. \left( 1 + \left[ \prod_{k=1}^n r(x_k)^{\theta_{M-1,k}} \right] \left[ \prod_{k=1}^n x_k^{\theta_{M,k}} \right] \right) \right)$$

$$\left. \left. \cdots \right) \right) \right)$$

is an approximation to $g_1$ in $K$, to within $\varepsilon/3$, with respect to $\|\cdot\|_\infty$. This is possible by equations (C.1) and (C.2); in particular the approximation should be sufficiently precise that

$$r^{2M-2}([(\sup K)^{-1}, \sup K]) \subseteq L,$$

which is why $\alpha > 0$ is needed: note how $r^2$, and thus $r^4$, $r^6$, ..., $r^{2M-2}$, are approximately the identity function on $L$.

This description of $\widetilde{g}_1$ is now amenable to representation with a neural network. The key fact about this description of $\widetilde{g}_1$ is that, working from the most nested set of brackets outwards, the value of $\widetilde{g}_1$ may be computed by performing a single chain of multiplications and additions, and occasionally taking the reciprocal of all of the input values.

So let the computation neuron and the out-register neuron corresponding to $g_1$ perform the multiplications, layer-by-layer, to compute $\prod_{k=1}^n x_k^{\theta_{M,k}}$, in the manner of Lemma C.1. Store this value in the out-register neuron.

Now use the computation neurons and the in-register neurons corresponding to $x_1$ (across multiple layers) to compute $r(x_1)$, in the manner of Proposition C.3: eventually the in-register neuron is now storing $r(x_1) \approx 1/x_1$. Repeat for the other in-register neurons, so that they are collectively storing $r(x_1), \ldots, r(x_n)$.

Now the computation neuron and the out-register neuron may start multipying $r(x_1), \ldots, r(x_n)$ on to $\prod_{k=1}^{n} x_k^{\theta_{M,k}}$ (which is the value presently stored in the out-register neuron) the appropriate number to times to compute $\left[\prod_{k=1}^{n} r(x_k)^{\theta_{M-1,k}}\right] \left[\prod_{k=1}^{n} x_k^{\theta_{M,k}}\right]$, by Lemma C.1. Store this value in the out-register neuron. Then add one (using the affine part of a layer). The out-register neuron has now computed the expression in the innermost bracket in the description of $\widetilde{g}_1$.

The general pattern is now clear: apply $r$ to all of the in-register neurons again to compute $r^2(x_i)$, multiply them on to the value in the out-register neuron, and so on. Eventually the out-register neuron corresponding to $g_1$ will have computed the value $\widetilde{g}_1$. Actually, it will have computed an approximation $\widehat{g}_1$ to this value, because of the identity approximations involved.

By taking all of the (many) identity approximations throughout the network to be suitably precise, the values of $\widetilde{g}_1$ and $\widehat{g}_1$ may be taken within $\varepsilon/3$ of each other, and the values of $\widehat{g}_2, \ldots, \widehat{g}_m$ and $g_2, \ldots, g_m$ may be taken within $2\varepsilon/3$ of each other, in each case with respect to $\|\cdot\|_\infty$ on $K$.

Thus $(\widehat{g}_1, \ldots, \widehat{g}_m)$ approximates $f$ with total error no more than $\varepsilon$, and the proof is complete. $\qquad\square$

