# OpenReview forum: "Universal Approximation with Deep Narrow Networks"
_ICLR.cc/2020/Conference — Reject_

### Official Review · AnonReviewer2 · 2019-10-23
**Official Blind Review #2**

**Rating:** 8

**Review:**

This paper proves universal approximation theorems for fully connected networks with fixed width and unbounded depth. Unlike recent results focusing on ReLU networks approximating scalar valued target functions, this paper provides universal approximation theorems for a wide range of activation functions and vector valued target functions.

This paper provides a number of universal approximation theorems on different activation functions, but the central result can be stated as follows:
Given input dimension n and output dimension m and activation \rho, assume \rho is a continuous function and there exists \alpha \in R such that \rho is continuously differentiable at \alpha and the derivative \rho’ is nonzero at \alpha. Then, a fully connected network with layer width n+m+2 and unbounded depth can approximate any continuous function on a compact domain to arbitrary sup-norm accuracy.

I would like to vote for acceptance of this paper because the authors develop nontrivial techniques that extend existing universal approximation results on width-bounded ReLU networks to essentially “all” other activation functions. I think this paper is well-written and well-organized; it gives a good overview of the existing results that contextualizes this paper well, briefly summarizes the main results, and then reveals the details of construction. I haven’t checked the full details of the proofs in the appendix, but as far as I can tell they look correct.

While I think that Section 4 reveals the proof techniques reasonably well, I believe that proofs of Propositions 4.3 and 4.7 should be covered/sketched in greater detail in the main text. For example, Theorem 4.4 builds upon Proposition 4.3 by approximating identity function locally using \rho. The main text only reveals approximation of identity function by \rho and defers the whole proof of Proposition 4.3 to the appendix. I think adding proof sketches for the propositions will be more helpful to the readers, and to this end, cutting down some text in the recurring paragraph “uniform continuity preserves uniform convergence…” should be helpful.

According to Remark 4.9, it seems that the proof strategy for polynomials can also be applied to nonpolynomials. This motivates a natural question: why can’t you apply the proof strategy for nonpolynomials to polynomials? I believe you can’t because Proposition 4.3 relies on the existing universal approximation results which requires \rho to be nonpolynomial (which is not mentioned in the main text). In my opinion, adding some comments on “why nonpoly techniques can’t be applied to poly activations” would help readers better understand the proof techniques.

Another question just out of curiosity: can you prove tightness of your construction, e.g., show that a square model with width n+m is NOT a universal approximator?

**Experience Assessment:**

I have published one or two papers in this area.

**Review Assessment: Checking Correctness Of Derivations And Theory:**

I assessed the sensibility of the derivations and theory.

**Review Assessment: Checking Correctness Of Experiments:**

N/A

**Review Assessment: Thoroughness In Paper Reading:**

I read the paper at least twice and used my best judgement in assessing the paper.

---

> ### Author Response · Authors · 2019-11-06
> **Author Response**
>
>
> Thankyou for your time and for your feedback.
>
> We agree that it would be nice to sketch the proofs of Propositions 4.3 and 4.7 in the main text. We had already aimed to do this; in response to this feedback we adjusted these to add a little more detail, in particular for Proposition 4.7.
>
> Regarding applying the proof strategy for nonpolynomials to polynomials: you are entirely correct that this can't be done, due to the use of the existing univeral approximation results. We have added some extra text to Remark 4.9 to mention this.
>
> In response to the question about tightness: that's a tough question! Lower bounds are much harder to come by, and we do not have better lower bounds than are already known. To the best of our knowledge, the best known lower bounds are in Johnson 2019, and Hanin and Sellke 2017, but neither one provides lower bounds for the case of vector-valued target functions (i.e. there is dependence on $n$ only).
>
> Thankyou once again for your review.

---

> > ### Comment · AnonReviewer2 · 2019-11-14
> > **Author Response Acknowledged**
> >
> > Dear Authors,
> >
> > I have read your response as well as the updated manuscript. Thanks for your efforts.

---

### Official Review · AnonReviewer1 · 2019-10-24
**Official Blind Review #1**

**Rating:** 6

**Review:**

    - This paper complement of the fundamental Universal approximation theorem variants
    - Based of the Register model, that the authors seem to have developed themselves from the scratch, elegant, although non-obvious
    - Proof is straightforward, although the pictural description can be enhanced. In reality the width neurons are unfolded horizontally in the n+m+1 layer
    - As of now, single-point continuous function does not seem to have been proven to be sufficient to build universal single-layer approximator networks in the 1999 paper.
It is unclear how the authors prove that part of their theorems.
    - The third part of the paper relies on Stone-Weierstrass theorem and a manipulations around the concept of "enhanced neurons", carefully constrcutred to fit in the n+m+1 and n+m+2 budgets
    - However, the proof of relaxing the Polynomial functional constraint in Theorem 4.8 is not entirely clear. While it seems to be a two-staged proof (convergence for non-linear function x^2, then convergence of a class of polynomial functions to x^2), it is unclear how the $\rho_h$ neurons can be assembled with the registry neuron budget from the initial polynomial function.
    - Although inspired by prior work, the author's contribution is novel, original and important.

Overall, I find this paper highly useful, elegant and, to the extent of my knowledge and understanding, properly proved. My main suggestions to the authors are with regards to the clarification of the proof of the Theorem 4.8.  after which I could increase my score.

Acknowledging the rebuttal: thank you for your clarification and for updating the paper accordingly.


**Experience Assessment:**

I have published one or two papers in this area.

**Review Assessment: Checking Correctness Of Derivations And Theory:**

I assessed the sensibility of the derivations and theory.

**Review Assessment: Checking Correctness Of Experiments:**

N/A

**Review Assessment: Thoroughness In Paper Reading:**

I read the paper at least twice and used my best judgement in assessing the paper.

---

> ### Author Response · Authors · 2019-11-06
> **Author Response**
>
>
> Thankyou for your time and your feedback.
>
> You note a concern about whether there are existing results for universal approximation with "single-point continuous function". However we are assuming that our activation functions are continuous everywhere. We do have a condition at a single point, but that is a condition on continuous differentiablity (that is, that the derivative exists, and that the derivative is also continuous); this is not a condition on the continuity of the activation function itself.
>
> You note that it is particularly important to try and clarify the proof of Theorem 4.8, with regard to how the $\rho_h$ neurons are used. We have adjusted the final paragraph of the proof of Theorem 4.8; hopefully this improves on any lack of clarity.
>
> Thankyou again for your review.

---

### Official Review · AnonReviewer3 · 2019-10-28
**Official Blind Review #3**

**Rating:** 3

**Review:**

Summary: The paper presents approximation power results for deep, but narrow networks with various activation functions. In particular, the authors target the minimum width possible, s.t. the class of networks considered remain universal approximators. The authors consider ReLU activation functions, polynomial activations, as well as some non-differentiable activations.

Evaluation: I'm personally not very closely involved with some of the recent developments on representational power of narrow, deep networks, but it seems to me this paper follows mathematically similar intuitions to prior works (e.g. "memorize" the input / prior computations; another way to visually represent these proofs is to "rotate 90 degrees" the usual shallow network approximation -- the authors call this a "register" model. They cite the paper by Lu et al '17 which essentially uses this proof technique).

While there are some technically interesting parts (e.g. the polynomial activation part has some trickiness in maintaining the width at n+m+1), I don't think will be very interesting to the ICLR community at large, and I think it is fairly incremental.

The writing is by and large clear, though Appendix C is a little wordy and hand-wavy.



**Experience Assessment:**

I have published one or two papers in this area.

**Review Assessment: Checking Correctness Of Derivations And Theory:**

I assessed the sensibility of the derivations and theory.

**Review Assessment: Checking Correctness Of Experiments:**

N/A

**Review Assessment: Thoroughness In Paper Reading:**

I read the paper at least twice and used my best judgement in assessing the paper.

---

> ### Author Response · Authors · 2019-11-06
> **Author Response**
>
>
> Thankyou for your time and your feedback.
>
> You note that "The authors consider ReLU activation functions, polynomial activations, as well as some non-differentiable activations". We would like to highlight that in fact we consider a much broader class of activation functions than these. We belive that an important part of this work is to present results for almost any activation function.
>
> We agree that memorization is used in previous work, but this seems to be intrinsic to the nature of the problem. Width limited neural networks may be thought of as memory-limited algorithms; precisely what is kept in memory must be carefully managed.
>
> Finally we acknowledge that Appendix C is a little wordy - we found that describing the same setup in mathematical notation simply became an intractable mass of symbols! By contrast, consider the proof of the much simpler Register model, in Figure 3: this is already at the limit of what can reasonably be followed.
>
> Once again, thankyou for your review.

---

> > ### Comment · AnonReviewer3 · 2019-11-14
> > **Acknowledging response**
> >
> > Dear authors,
> >
> > I have read and acknowledge your response. Thank you for taking the time to respond -- my opinion remains the same.

---

### Decision · Program_Chairs · 2019-12-19

**Decision:**

Reject

**Comment:**

This article studies universal approximation with deep narrow networks, targeting the minimum width. The central contribution is described as providing results for general activation functions. The technique is described as straightforward, but robust enough to handle a variety of activation functions. The reviewers found the method elegant. The most positive position was that the article develops non trivial techniques that extend existing universal approximation results for deep narrow networks to essentially all activation functions. However, the reviewers also expressed reservations mentioning that the results could be on the incremental side, with derivations similar to previous works, and possibly of limited interest. In all, the article makes a reasonable theoretical contribution to the analysis of deep narrow neural networks. Although this is a reasonably good article, it is not good enough, given the very high acceptance bar for this year's ICLR.